# Agrivoltaic Systems Enhance Farmers' Profits through Broccoli Visual Quality and Electricity Production without Dramatic Changes in Yield, Antioxidant Capacity, and Glucosinolates

**Seung-Hun Chae [1,2,†], Hye Joung Kim [1,†], Hyeon-Woo Moon [1,2], Yoon Hyung Kim [3] and Kang-Mo Ku [1,2,*]**

[1] Department of Horticulture, College of Agriculture and Life Sciences, Chonnam National University, Gwangju 61886, Korea; 216544@jnu.ac.kr (S.-H.C.); sahm@jnu.ac.kr (H.J.K.); 218845@jnu.ac.kr (H.-W.M.)

[2] Interdisciplinary Program in IT-Bio Convergence System at Chonnam National University, Gwangju 61186, Korea

[3] Department of Agricultural Economics, College of Agriculture and Life Sciences, Chonnam National University, Gwangju 61886, Korea; yonhk@jnu.ac.kr

[*] Correspondence: ku9@jnu.ac.kr; Tel.: +82-62-530-2065

[†] These authors contributed equally to this work.

**Abstract:** The increase in world population by an average rate of 2% per year causes critical issues on energy and foods. By 2050, food demand will increase to 35~56% more than in 2010 due to the growing population. Agrivoltaic systems allow us to reach sustainable food and electricity-production goals with high land-use efficiency. In this study, the yield, antioxidant capacity, and secondary metabolite of broccoli and electricity production were analyzed under an agrivoltaic system over 3 cultivation periods. Based on energy production, an economic analysis of agrivoltaic was carried out. In addition, our study also reported that agrivoltaic with additional shading treatment produced greener broccoli with a higher level of consumer preference than open-field grown ones. The yield, antioxidant capacity, some glucosinolates and hydrolysis products of broccoli grown under an agrivoltaic system were not significantly different from those of broccoli grown in the open-field.

**Keywords:** broccoli; agrivoltaic; solar panel; shading; quality; glucosinolates

## 1. Introduction

Over the past few decades, the consumption of energy resources has increased due to the growing world population [1], which increases by an average rate of 2% per year and causes critical issues for the environment [2]. By 2050, food demand will increase to 35~56% more than in 2010 due to the growing population. Traditional agriculture uses large amounts of heating, fertilizers, and various agricultural machinery to increase yields. This excessive use causes a large amount of energy consumption and high $CO_2$ emissions. Efforts to reduce greenhouse gas and carbon emissions in the energy sector, which includes power generation using fossil fuels, are urgently required as these emissions have emerged as a national issue. To solve this problem, the utilization of renewable energy has been increasing quickly, but according to International Energy Agency (IEA) statistics, South Korea's renewable electricity output remains lowest among OECD countries. This is because Korea has limits to expanding its renewable energy industry because its land area is relatively small and 70% of it is covered with mountains. Additionally, while large-scale renewable energy production for national energy demands must be met, it can simultaneously cause problems for the environment, i.e., soil erosion or landslides.

To solve these problems, an agrivoltaic (AV) system that can facilitate both farming and renewable energy generation by erecting photovoltaic panels on crop land has received attention [3]. This system can not only grow crop plants but can also generate

renewable energy to increase land use efficiency and secure farmers' income as it allows them to use their own agricultural land for two purposes. One way to improve AV efficiency is by utilizing a bifacial photovoltaic panel, thereby capturing photons from both sides. In fact, a recent study has reported that bifacial photovoltaic solar panels have a 20% higher capacity for generating electricity than mono-facial panels [4].

Typically, solar panels are installed 3–4 m above ground to make space for tractors or other agricultural machines in the AV system (Figure 1A). They are also tilted at a 30° angle to efficiently capture sunlight to the south. Due to the space between solar panels, some crops can capture sunlight directly from the sun, while other crops can be placed in the solar panels' shade. This shaded area moves as the sun moves over the course of the day (Figure 1B). Given the growing environment under the AV system, crops will receive less sunlight than those grown without AV systems. Generally, plants with higher light saturation points can grow well under intense light, but plants with lower light saturation points can grow relatively well in the shade [5]. Each crop has different light saturation points, indicating that certain species or plant families may have better shade tolerance under solar panel shading. Plants such as tomatoes, eggplants, and cucumbers have 45–70 Klux light saturation points, while crucifers such as broccolis, cabbages, and Kimchi cabbages have 11 Klux [6]. Among the greenish-yellow plants, broccoli is nutritious, containing vitamin C, polyphenols, carotenoids, and tocopherols, as well as high levels of other healthy phytonutrients [7,8]. Since it has 4 or 27 times more vitamin C than cabbage or lettuce, respectively, it was ranked as a top nutritional plant among 16 vegetables in a vegetable nutritional evaluation in the United States [9]. In particular, the glucosinolates in broccoli are known as phase II enzyme inducers (detoxifying enzymes) or anticancer compounds, which fight against bladder, breast, and liver cancers [10–12]. To provide crop quality information under an agrivoltaic panel, glucosinolates and antioxidant activity of broccoli in the AV system can be important information for consumers.

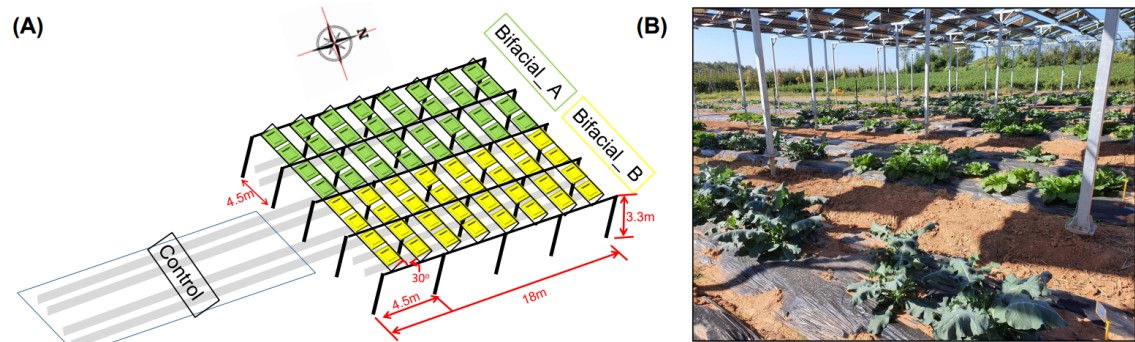

**Figure 1.** Agrivoltaic structure information for this experiment (**A**) and photo of growing crops under the solar panel (**B**).

Because of its low light saturation points, broccoli may be a suitable crop to maximize famer's profits and energy security through the AV system. However, to date, there is limited information on the performance of brassica crops in the AV system. Thus, this study attempts to determine the quality and economic feasibility of broccoli farming during fall (2019), spring (2020), and fall (2020) periods.

In the case of broccoli head color, it is green, purple, and dark purple depending on the cultivation environment and genotype. Topcu et al. [13]. and Garitta et al. [14]. reported that consumers preferred green-colored broccoli heads. In terms of varieties, green varieties were preferred to purple varieties in local markets and supermarkets [15,16]. Accordingly, the color of broccoli is an important property that goes beyond appearance quality and is involved in consumers' desire to purchase. However, it is important to note that broccoli has never been studied as a crop plant for a bifacial photovoltaic panel system in terms of visual quality enhancement and health-promoting phytochemicals. Due

to the partial shading by AV, we hypothesize that broccoli would show a greener color than conventional agronomic practice and contain different levels of phytochemicals and yields. It is possible that broccoli production may be more profitable under AV systems if an active shading treatment is applied by utilizing the AV structure. Thus, we evaluated additional shading treatment effects on broccoli head colors as well as agronomic and economic aspects of broccoli production in the AV system.

## 2. Materials and Methods

### 2.1. Broccoli Cultivation under Agrivoltaic Systems and Shading Treatment

An AV system was established in Naju, Jeollanam Province, South Korea (34°58′02.4″ N, 126°45′55.9″ E) in July 2019. Our study period was from the fall of 2019 to the spring and fall of 2020. Five weeks after germination, the seedlings were moved outside for hardening off. The plants were grown in the greenhouse at Chonnam National University under a night- (18 °C) and day- (30 °C) time air temperature regime. The broccoli cultivar was transplanted with its proper planting space (45 cm × 45 cm). The experiment design was a random complete block design with 4 replications. Each replication area had 20 broccoli plants. For comparison, plants were also grown without solar panel structures as the control group. The control group was placed south of the AV system to avoid a shade effect from the solar panel and to match similar soil properties. Conventional agricultural practices were applied as needed. Throughout the experiment, the growth status of each crop was inspected. Furthermore, broccoli florets were harvested with an 8 cm stem from each broccoli plant at commercial maturity. The harvested broccoli florets were then freeze-dried, ground to a powder, and kept at −20 °C.

For an additional experiment to utilize the AV structure for shading treatment, broccoli was cultivated from 26 July to 23 November 2021. On 17 October 2021, broccoli florets started to emerge, and a 35% shading net was installed on the plants (Figure 2A). In our study, some broccoli plants in north AV were removed at growth parameters and analysis due to the environment, for example, open-field (Figure 2B). Shading nets were installed using a bracket against an AV pillar (Figure 2C). The beds were not covered with black plastic mulching film to reduce carbon emissions. After harvest, the weight of each broccoli head was measured immediately by scale.

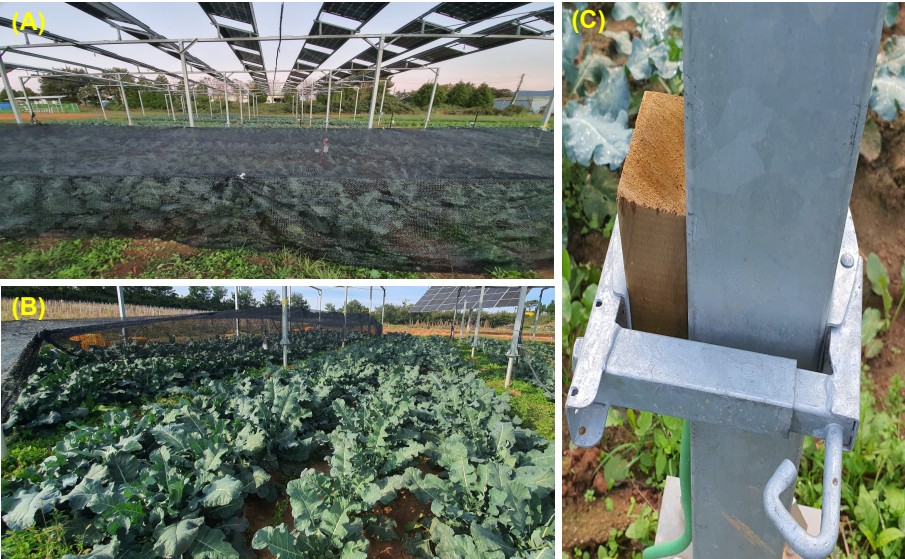

**Figure 2.** Photos of the west (**A**) and south side (**B**) agrivoltaic structure with additional shading in this experiment and bracket for installing shading net (**C**).

### 2.2. Plant Growth Parameters

The height of a broccoli plant, from the ground to the broccoli head, was measured prior to harvesting. Meanwhile, the weight of the plants was measured after harvesting five broccoli florets by cutting the stems uniformly at 8 cm and weighing their heads. The thickness of the stem was measured using calipers just below the head. Broccoli weight and height were collected from five broccoli for one biological replication for four experiment blocks.

### 2.3. Sample Extraction

A total of 75 mg of freeze-dried powder of each sample was extracted with 1.5 mL of 80% ethanol at 90 °C for 10 min. After 5 min of cooling on ice, the extract was centrifuged at $10,000 \times g$ for 5 min. The supernatants were then transferred to a 1.5 mL microcentrifuge tube. This extract was used for 2,2-diphenyl-1-picrylhydrazyl (DPPH) antioxidant activity assays and to quantify the total phenolic and flavonoid concentrations based on the dried tissue weight.

### 2.4. Total Phenolic Content (TPC)

Total phenolic content and antioxidant capacity were analyzed following a previously published method with minor modifications [17]. Each sample (20 μL) was mixed with 100 μL of Folin–Ciocalteu reagent (0.2 N), followed by 3 min of incubation at room temperature. Subsequently, 80 μL of sodium carbonate (7.5%) was added. After 60 min of incubation in the dark at room temperature, absorbance was measured at 750 nm using a spectrophotometer (SpectraMax® ABS Plus, Molecular Devices, San Jose, CA, USA). The total phenolic concentration was determined based on a standard curve of gallic acid.

### 2.5. Total Flavonoid Content (TFC)

To measure total flavonoid content, each sample (20 μL) was thoroughly mixed with diethyleneglycol for 3 min, followed by the addition of NaOH (10 μL, 1 N) for 1 h at 37 °C to allow for a proper reaction. A standard curve was drawn by measuring absorbance at 427 nm using a reference, naringin; total flavonoid content was expressed as naringin (mg) per dry weight (g).

### 2.6. Determination of the Antioxidant Activity by the 2,2-Diphenyl-1-Picrylhydrazyl (DPPH) Free-Radical Scavenging Assay

The DPPH assay was conducted as described by Ku et al. [17] with minor modifications. Reaction mixtures containing test samples (10 μL) and 190 μL of a 200 μM DPPH ethanol solution were incubated at room temperature for 30 min in 96-well plates. The absorbance of the DPPH free radicals was measured at 517 nm using a SpectraMax® ABS Plus microplate reader (Molecular Devices). Results are expressed as a percentage of scavenging activity compared to the control.

### 2.7. Quantitation of Glucosinolate

Glucosinolates were analyzed following the methods described by previous publications [18,19] with minor modifications. Freeze-dried broccoli powder from each sample (0.2 g) was weighed into a 15 mL conical tube and mixed with 2 mL of 70% methanol. After heating at 95 °C for 10 min in a heating block, the tubes were cooled on ice for 5 min before the addition of an internal standard (sinigrin). The tubes were then vortexed and centrifuged at $12,000 \times g$ for 10 min at room temperature, and the supernatants were collected. The pellets were then extracted again with 2 mL of 70% methanol. Protein in the pooled extract was removed by mixing with 0.15 mL of a mixture of 1 M lead acetate and 1 M barium acetate (1:1, *v/v*). Desulfo-glucosinolates were prepared using Sephadex A-25 resin. The filtered sample was injected into an HPLC for glucosinolate quantification [20,21].

### 2.8. Glucosinolate Hydrolysis Products

The glucosinolate hydrolysis products were analyzed using gas chromatography–mass spectrometry (GC-MS) following the method described by Kim et al. [22].

### 2.9. Color Characteristic Measurements

After harvesting, five parameters (L*: lightness, a*: redness, b*: yellowness, C*: chroma, Hue angle) were measured using a colorimeter (NR60CP, Shenzhen 3nh Technology Co., Ltd., Shenzhen, China) for comparing the color of broccoli heads among those grown from OP, AV, and AV plus 35% shading (SD). Each treatment had seven biological replicates.

### 2.10. Microclimatological Measurements

Four microclimates and soil environments were measured in the control area and solar panel area. Measured data were air temperature (ATMOS14, Meter, Pullman, WA, USA), water content (ATMOS14, Meter, Pullman, WA, USA), soil temperature (TEROS11, Meter, Pullman, WA, USA), and photosynthetically available radiation (Photosynthetic Photon Flux Density; PPFD, PAR, Meter, Pullman, WA, USA). Data were logged on ZL6 loggers (ZL6, Meter, Pullman, WA, USA).

Soil temperature (TEROS11) was measured at two spots within 7.5 cm of the surface. Air temperature and water content (ATMOS14) were measured at two spots above 100 cm of the surface. Microclimate and soil environment were measured in 10 min intervals. Graphs were used by editing ZENTRA Cloud (ZENTRA, Meter, Pullman, WA, USA).

### 2.11. Electricity Generation and Economic Evaluation of Solar Panel

Electricity generation was reported by using total power plant diagnostic system (www.solar.mrt.co.kr, accessed on 20 January 2011). When using solar power generation, there may be two kinds of benefits: private and public benefits. Private benefits refer to those from reduced annual electricity cost for farmers who have used existing electricity for agricultural purposes, and public benefits mean the loss reduction from KEPCO's agricultural electricity sales and $CO_2$ reduction benefits.

Annual amount of power generation (kWh) = Average operating hours per day (hr) X Average power generation per day (kWh) × Days of operation (day)

Amount of annual cost reduction (kWh) = Amount of electricity saved for farmers (kWh) × Unit cost of electricity for agricultural purposes (kWh)

Benefits of annual reduction (USD) = Amount of electricity savings in agricultural use (kWh) × (System marginal price, SMP—Electricity price for agriculture)

Amount of reduction (USD) = Amount of electricity savings in agricultural use (kWh) X Emission coefficient ($tCO_2$/kWh) × Emission permit price (USD/$tCO_2$)

The annual economic benefit from solar power use is determined by the sum of the amount of annual cost reduction, benefits of annual reduction, and amount of reduction. The dollar exchange rate was on average 1184.27 during the experimental period.

### 2.12. Statistical Analysis

Statistical significance for all the experimental results was determined using SPSS (Statistical Package for Social Science, version 26, SPSS Inc., Chicago, IL, USA). To test any standard differences between experimental treatments, the analysis of variance (ANOVA) was applied. Once there was statistical significance, the similarity among the treatments was estimated using Tukey's HSD test. Any significance for values of means was tested within a confidence level of $p < 0.05$.

## 3. Results and Discussion

### 3.1. Plant Growth Environment

'*Early You*' F1 Broccoli seedlings were transplanted when they were about 30 days old and had five to six true leaves. The average number of days from transplanting to harvesting (DTH) of fall 2019 (from 26 August to 18 November 2019), spring 2020 (from 28 April 28 to 1 June 2020), and fall 2020 (from 6 September to 21 November 2020) was 74 days, 34 days, and 75 days, respectively. Additionally, the average growing degree day (GDD) of broccolis during fall 2019, spring 2020, and fall 2020 from transplanting to harvest was 877.5 °C/day, 434.8 °C/day, and 645.7 °C/day, respectively. The averages of solar radiation in the fall of 2019, spring of 2020, and fall of 2020 were 3562 Wh/m², 5553 Wh/m², and 3399 Wh/m², respectively. The average precipitation amounts for each season were 388 mm, 199 mm, and 41 mm, respectively (Table 1). Average DTH and solar radiation of five broccoli cultivars during 2009 fall and 2010 fall were 71.8 days and 1926 Wh/m², respectively [23]. AV was not different compared to the DTH of a previous study and open-field. Soil temperature and PPFD showed a difference between AV and control (Table 2). Average soil temperature of open-field (20.2 °C) was significantly higher than AV (19.1 °C). As of 27 September, soil temperature of open-field was 27.4 °C, soil temperature of AV was 22.3 °C. Soil temperature of open-field was 2 to 5 °C higher than soil temperature of AV. As of 28 September, the PPFD of open-field was 918 $\mu$mole·m$^{-2}$·s$^{-1}$, the PPFD of AV was 297.3 $\mu$mole·m$^{-2}$·s$^{-1}$. The PPFD of open-field was about three times higher than the PPFD of AV. For daily change of PPFD, the PPFD was lower than light saturation in the shade of the solar panel, but was higher than light saturation with no shade (Figure 3). Therefore, a unique environment in which the PPFD fluctuates was created by the shade of solar panel.

**Table 1.** The DTH (days from transplanting to harvesting), GDD (average growing degree day), solar radiation, and precipitation amount from open-field and agrivoltaic systems during the periods of fall 2019, spring 2020, and fall 2020.

| OF/AV | DTH | GDD (°C) | Solar Radiation (Wh/m²) | Precipitation (mm) |
|---|---|---|---|---|
| 2019 Fall | 79/74 | 902/878 | 3562 | 388 |
| 2020 Spring | 34/34 | 435/435 | 5553 | 199 |
| 2020 Fall | 74/74 | 646/646 | 3399 | 41 |

**Table 2.** Soil temperature (°C) and photosynthetic active radiation (PPFD, $\mu$mole·m$^{-2}$·s$^{-1}$) from open-field and agrivoltaic systems.

| | Open-Field | Agrivoltaic |
|---|---|---|
| Soil temperature | 20.2 ± 5.1 *** | 19.1 ± 4.1 |
| PPFD | 635 ± 59 *** | 369 ± 60 |

Microclimate and soil environment were measured in 10 min intervals. The value indicated daily average soil temperature and PPFD in all seasons. The values represent the mean ± standard deviation of three biological replications. *** indicates significant difference between AV and open-field by unpaired *t*-test ($p < 0.0001$).

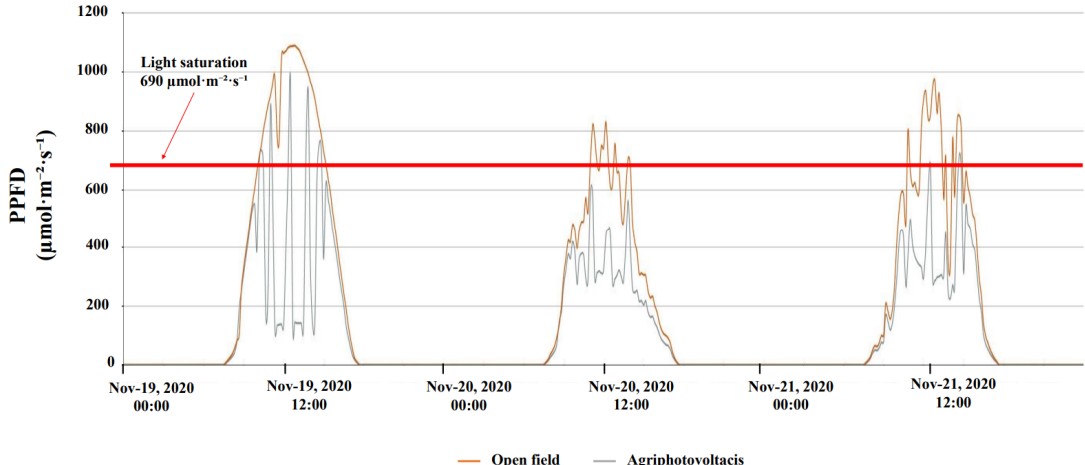

**Figure 3.** Daily Photosynthetic Photon Flux Density (PPFD) between open-field and agrivoltaic systems (10 min intervals data points). The graph represents typical daily PPFD during three seasons. Red line indicates the estimated light saturation point of broccoli.

*3.2. Characteristic of Growing Parameter*

The growing parameter of a broccoli during each cultivation season was measured (Figure 4). Depending on panel types, the average weight of the control broccoli head from three seasons was 305.8 g, whereas that of Bif A and B was 290.4 g (5% lower than control) and 265 g (13% lower than control), respectively (Figure 4A). There was no significant difference between different growing conditions (control vs. under the solar panel; F = 1.633, *p* = 0.212). The broccoli head grown in the fall period of 2020 weighed 409.6 g on average, which is significantly higher than the 2019 fall (279.8 g on average) and 2019 spring (178.1 g on average) broccoli heads. The drought stress in the fall period of 2020 was very severe. Thus, we actively irrigated both groups (control and under the solar panel) to maintain about 30% of soil water content. As a result, the average weight of broccoli head in the 2020 fall was 32% and 57% higher (F = 49.85, *p* = 0.000) than that of the 2019 fall and 2020 spring, respectively. Significant interactions between season and growing conditions was not observed (F = 0.944, *p* = 0.452). Jones-Baumgardt et al. [24] reported that dry weights were increased by 100% in four Brassicaceae crops, such as cabbage and mustard, depending on increased PPFD. Thus, the weight of Brassicaceae was known to be effected by PPFD. Due to the unique AV environment, the average weight of broccoli heads was not significantly different between the open-field and solar panels. Under the AV panel, broccoli does not continuously receive a lower light compared with the light saturation; broccoli seems to show no difference in weight due to receiving higher light than light saturation at regular intervals according to the movement of the sun.

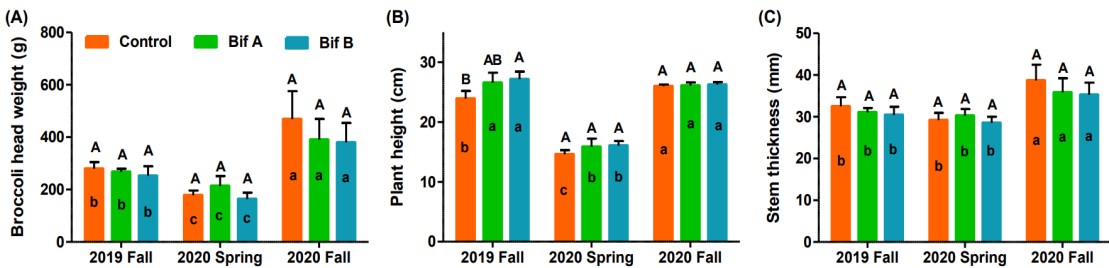

**Figure 4.** The average broccoli head weight (**A**), plant height (**B**), and stem thickness (**C**) of broccoli grown with or without solar panels. The control group indicates open-field growing conditions without a solar panel while BiF A and B indicate broccoli growing conditions under the bi-facial solar panels. The bars and error bars represent the mean ± standard deviation of three biological replications. Capital letters above the bars represent significant differences between different panels

in the same season. Small letters in the bars represent significant differences between different seasons in the same panel according to one-way ANOVA (Tukey's HSD test, $p < 0.05$).

The heights of broccoli grown in Bif A (26.6 cm) and B (27.1 cm) were significantly greater than the control (23.9 cm) in the 2019 fall season (Figure 4B). However, the trend was not consistent in the two consecutive seasons. The average stem thickness of broccoli grown in 2020 fall was 36.6 mm, which was significantly thicker than in the 2019 fall (31.3 mm) and in 2020 spring (29.3 mm, F = 28.946, $p < 0.001$, Figure 4C). Significant interactions between season and growing conditions was not observed (F = 0.741, $p = 0.572$).

### 3.3. Antioxidant Compounds and Capacity (TPC, TFC, and DPPH)

The average of total phenolic contents from the spring 2020 and the falls of 2019 and 2020 broccolis was 4.6 mg GAE/g DW and 5.2 (both of each) mg GAE/g DW, respectively, which is 11.5% higher than the spring 2020 broccolis (Figure 5A). The total phenolics of Bif A during the fall 2019 period was 10% lower than the control with a statistical significance. However, no significance was found among treatment groups for the spring 2020 and fall 2020 broccolis. The total flavonoids of the spring or the falls of 2019 and 2020 broccolis was 2.5 or 1.9(both of each) mg NAE/g DW, respectively (Figure 5B). The total flavonoids of broccolis were not significantly different between open-field and two different panels during the experiment period. The DPPH antioxidative activities of broccolis in the spring or the falls of 2019 and 2020 were 0.9 or 0.6 (both of each) mg GAE/g DW, respectively (Figure 5C). Those of Bif B during the spring 2020 period were 0.79 mg GAE/g DW, 24% less than those of the control (0.98 mg GAE/g DW), which was not significantly different between open-field and Bif B. Those of Bif A and B during the fall 2020 had no statistically significant differences compared to the control. Kavga et al. [25] reported that pepper under a glass panel was significantly different in antioxidant ability; for example, Folin–Ciocalteu, FRAP, and ABTS, compared to pepper in an open-field. Alfredo et al. [26] reported that the total flavonoid content of broccoli grown in the spring was similar to that of our study, which was higher than that of the fall. In our study, antioxidant capacity and yield were significantly different between seasons, while there was no significant difference between open-field and solar panels; it seems that broccoli was more suitable for a lower PPFD environment under AV than pepper.

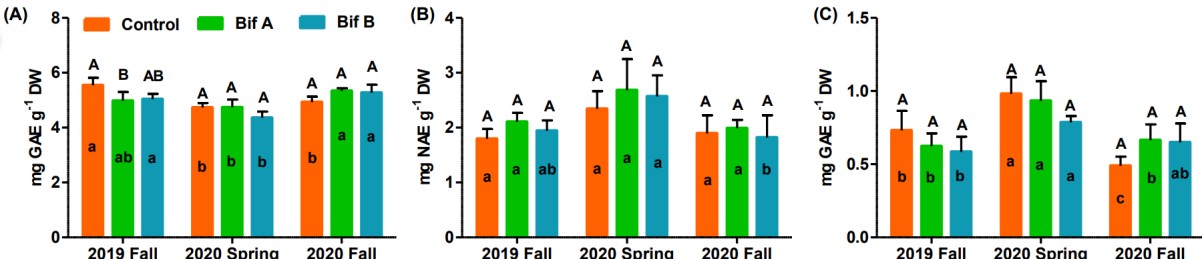

**Figure 5.** The total phenolics (**A**), flavonoids (**B**), and DPPH antioxidant activity (**C**) of broccoli grown with or without solar panels. The control group indicates open-field growing conditions without a solar panel, while BiF A and B indicate broccoli growing conditions under bifacial solar panels. The bars and error bars represent the mean ± standard deviation of three biological replications (seven broccoli florets were pooled for each replication). Capital letters above the bars represent significant differences between different panels in the same season. Small letters in the bars represent significant differences between different seasons in the same panel according to one-way ANOVA (Tukey's HSD test, $p < 0.05$).

### 3.4. Quantitation of Glucosinolates and Their Hydrolysis Products

Glucosinolates (GSL) are abundantly found in the cruciferous family, which includes cabbage, broccoli, and mustard plants. These possess various bioactivities including anticancer and anti-inflammatory activities in the human body. In addition, the metabolic

outputs of GSL have flavor-intensifying effects on vegetables. Therefore, the total GSL content and composition are important for developing crop varieties in the Brassica family for use as pest control means, anti-cancer drugs, flavor-intensifying agents, and so on. Glucoraphanin concentration of broccoli florets grown from fall 2019 was significantly higher than that grown in spring 2020 in control and BiF_B areas (Supplementary Figure S1A). This result is consistent with Farnham et al. [27] as they reported that glucoraphanin concentration was significantly correlated with days from transplant to harvest (DTH, 0.77**). Compared to DTH, the broccoli grown in fall 2019 had 74 DTH, while in spring 2020 there were 34 DTH. Farnham et al. [27] also reported that glucoraphanin correlated with quinone reductase induction potency per broccoli head (0.77***). Another study reported that glucoraphanin concentration was negatively correlated with photosynthetic photon flux [28]. Recent research also reported that broccoli grown in the fall season had a 6.25-fold higher amount of glucoraphanin than broccoli grown in the spring season [29]. Glucoraphanin accumulation is closely related to *AOP*2 gene expression. [30–32]. Thus, there may be a difference between *AOP* gene expression, DTH, and photosynthetic photon flux during the three different seasons. Glucobrassicin is a precursor of indole-3-carbinol, which is reported to possess various bioactive compounds, including some that have anti-obese [33], anticancer [34,35], and anti-extrogenic effects [36,37]. There was no significant difference in glucobrassicin concentration among treatments as well as seasons (Supplementary Figure S1C). It has been reported that neoclucobrassicin is a biomarker of insect damage or inducible compounds by jasmonic acid or methyl jasmonic acid [20,22,38–41]. The hydrolysis compounds of neoglucobrassicin were closely related with consumer preferences for broccoli [41]. Thus, neoglucobrassicin plays an essential role in broccoli quality. Generally, broccoli grown in spring 2020 had significantly higher neoglubrassicin concentrations than that grown in fall 2020 in the same panel (Supplementary Figure S1E). However, neoglucobrassicin concentration of broccoli grown in an open-field area in fall 2019 was only significantly different with solar panels (Supplementary Figure S1E). We found more caterpillars and insect damage in the control (open-filed) area than in AV areas in fall 2020. Insects seem to prefer to eat broccoli in an open-field area rather than in an AV area during cold temperatures and after washing pesticides (observation but no data) but we could not find the same phenomenon in fall 2020 due to pesticide washing by rain. Previous research reported that high temperatures or photosynthetic photon flux could upregulate indole glucosinolate [28]. Vallejo et al. [42] reported that differences in glucosinolate according to the season was affected by moisture deficiency rather than by high temperatures. However, based on our study and previous studies [43,44], it was found that not only moisture deficiency but also high temperature and light temperature were directly or indirectly related to differences in glucosinolate according to the season. Therefore, the higher neoglucobrassicin concentration of broccoli grown in spring 2020 may be related to higher solar radiation during broccoli cultivation and/or higher temperatures at harvesting time. Total glucosinolate did not show a certain pattern among treatments or between seasons (Figure 6A). It has been reported that glucosinolate concentration can be affected by abiotic and biotic factors [28], and the microenvironment changed by the solar panel structure synergistically affects glucosinolate concentration by interactions between seasonal growing conditions (biotic and abiotic factors) and microenvironment change by AV structure. We only observed a large seasonal variation between spring and fall, as previous studies reported [28,29], as well as contradictory data on total glucosinolates between two fall seasons. Sulforaphane, which had important anticarcinogenic effects [45], was induced by drought stress and increased in the fall more than in the spring due to drought stress [46,47]. In our study, sulforaphane was not significantly different during the experiment period except for the Bif A in fall 2020, during which season bif A was significantly lower than in the open-field (Supplementary Figure S2A). For total hydrolysis product, the content under the panel was decreased in the fall, which experienced dry conditions (Figure 6B). These differences were shown to have reduced the hydrolysis product due to less drought [48] by the shade of the solar panel.

Synthetically, yield, such as head weight, stem thickness, and stem height and secondary metabolite of broccoli, was not significantly different between open-field and AV; thus, producing broccoli in AV was not problematic.

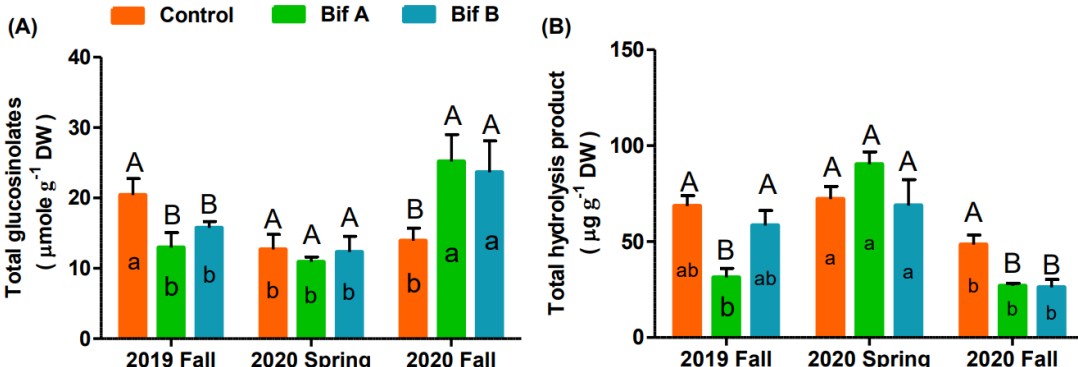

**Figure 6.** Total glucosinolate concentrations of broccoli grown with or without solar panels (**A**). Total glucosinolate hydrolysis product centration of broccoli grown with or without solar panels (**B**). Control indicates broccoli grown without solar panels, while BiF A and B indicate broccoli grown under solar panels. The bars and error bars represent the mean ± standard deviation of three biological replications. Capital letters above the bars represent significant differences between different panels in the same season. Small letters in the bars represent significant differences between different seasons in the same panel according to one-way ANOVA (Tukey's HSD test, $p < 0.05$).

### 3.5. Broccoli Visual Quality Change by AV Structure with Additional Shading

Under the AV structure, the broccoli color was greener than OF-grown broccoli. However, some gray-purple broccolis were observed in the southern area under AV (Figure 7). Because the southern area under the raised solar panel in AV was fully exposed to sunlight in the middle of the day, the light condition of this area was similar to OF. About 25% of broccoli under AV was still similar to open-field-grown broccoli.

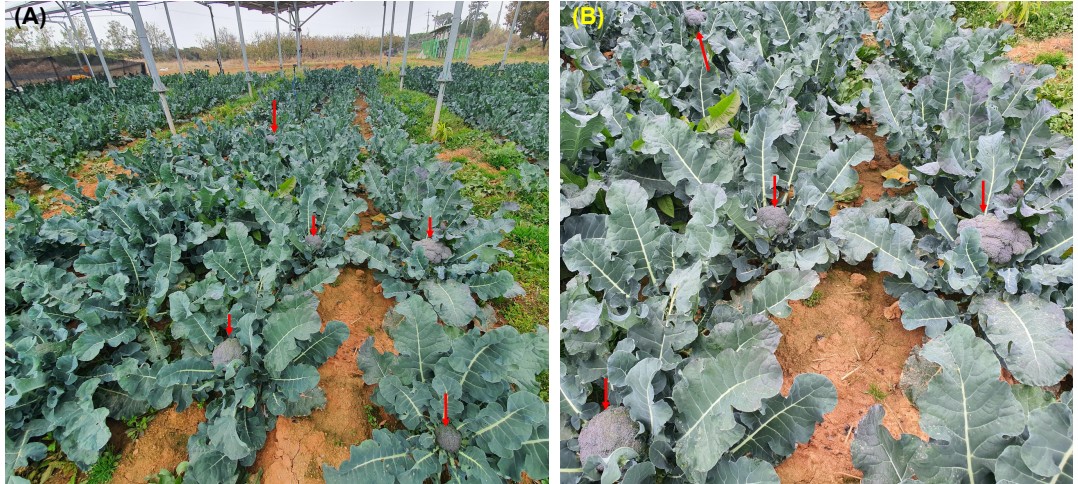

**Figure 7.** Photos of the south area under solar panel (**A**) and open-field grown broccoli (**B**) just before harvesting time. Red arrows indicate gray–purple broccoli florets.

For the production of uniform and preferred broccoli, an additional shading experiment was conducted. As a result of shading, visual quality and quantified color information showed differences (Figure 8A). The broccoli head color looks more purple grown from OF compared with the other two treatments. There was some broccoli that turned purple under AV, whereas all those that were shaded were green. The indications of color changed according to treatment (Figure 8B). The *L** was significantly higher for AV (36.5

± 3.5) and shading (38.9 ± 3.9) broccoli than the control (33.6 ± 3.3). The $a^*$ was significantly higher for the control (−1.1 ± 1.4) than AV (−5.1 ± 1.1) and AV was significantly higher than shading (−6.2 ± 1.7). The $b^*$ was significantly higher for shading (18.3 ± 2.8) than AV (15.1 ± 2.1) and AV was significantly higher than the control (9.1 ± 2.1). The $C^*$ was significantly higher for shading (19.3 ± 3.1) than AV (16.0 ± 2.2) and AV was significantly higher than the control (9.3 ± 2.1). The hue angle was significantly lower for AV (108.8 ± 3.3) and shading (109.0 ± 4.5) broccoli than the control (95.3 ± 8.7). The value of data was expressed as 'mean ± standard deviation'. According to the color space diagram, higher $L^*$ means brighter, lower $a^*$ means greener, and higher $b^*$ means more yellowish [49]. This is a comparison of broccoli grown from AV with the control and quantitatively shows that agricultural solar produced broccoli is closer to green. The average weight of each treatment was also measured (Figure 8C). The mean broccoli weight of control (427 ± 128) was significantly higher than AV (384 ± 84) and shading (358 ± 75) treatment ($p < 0.05$, $n = 100$). The yield reduction percentage of AV and shading compared to the control was 10.0 and 16.1%, respectively. Delayed harvests were also observed in treatments of this experiment. The harvest was started equally on 9 October 2021, yet on the first harvest day, 21% of the total was harvested in the control area, whereas 1% in AV and 2% were harvested in the shaded area. In addition, on 15 November 2021, harvesting was finished for the control, but harvesting was completed for AV and shading on 23 November 2021. During the shading period, the average PPFD for the control, AV, and shading were 532.6 μmol·m$^{-2}$·s$^{-1}$, 309.7 μmol·m$^{-2}$·s$^{-1}$, and 219.6 μmol·m$^{-2}$·s$^{-1}$, respectively. Labos et al. [50] showed that as the shading rate increased, the yield of blueberry (*Vaccinium corymbosum* var. Elliott) decreased and fruit maturation was delayed. Sophie et al. suggested that under shade conditions at the early flowering stage, there was a 43.4% reduction in photosynthetic active radiation (PAR) and the vegetative growth of rapeseed (*Brassica napus* L.) increased, whereas reproductive growth decreased [51]. The time until commercial maturity of broccoli increased from 35% to 70% from 5 days to 11 days by AV structure shading and an additional shading in AV, respectively. The proportion of dry matter of broccoli head decreased (6%) in 70% shading [52]. In our experiment, 58% and 41% PAR were reduced in AV and additional shading conditions, respectively. Therefore, the decrease in yield and the delayed harvest are likely due to a decrease in light.

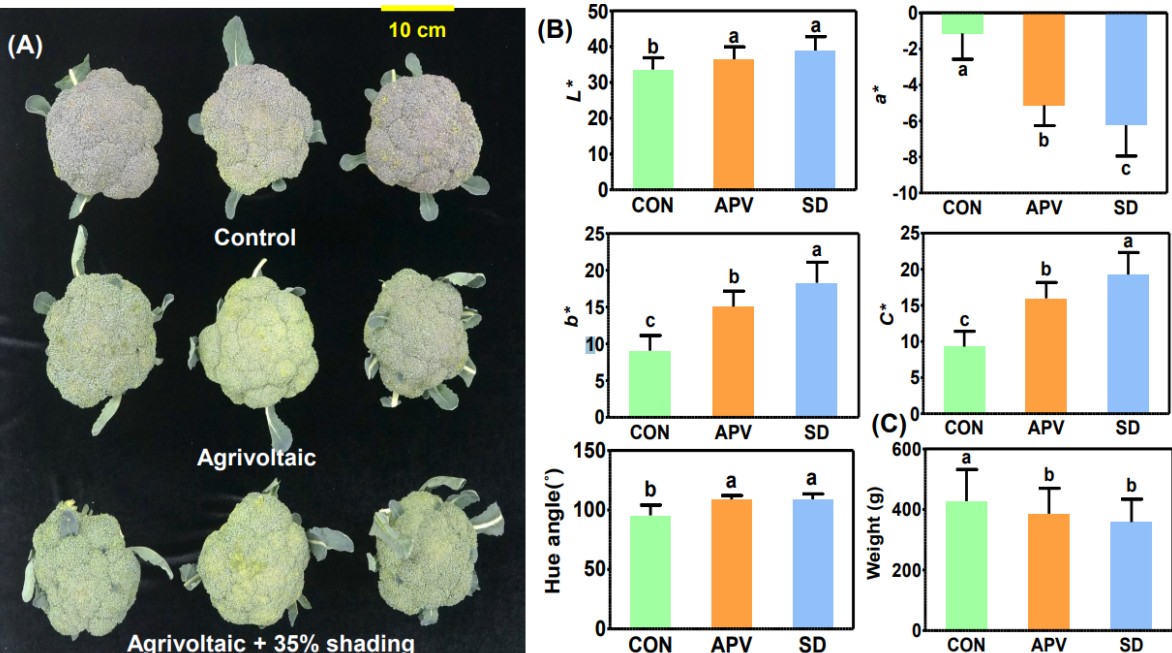

**Figure 8.** Photos representing each treatment (**A**), color characteristics ($L^*$: Lightness, $a^*$: redness, $b^*$: yellowness, $C^*$: chroma) according to each treatment (**B**), and average weight of broccoli (**C**). CON:

Control, AV: Agrivoltaic, SD: Agrivoltaic + 35% shading net. Different lowercase letters above error bars indicate significant difference among different treatments assessed by Tukey's HSD at $p = 0.05$. Error bars indicate standard deviation ($n = 20$ and 100, color characteristics and weight, respectively).

*3.6. Economic Evaluation of Solar Panel*

During the experiment period, the average operating hours per day was 4.2 h, the average power generation per day was 127 kWh, and the annual power generation was estimated to be 46,355 kWh (Supplementary Table S1A). When solar power is used, the annual electricity savings for farming are 46,355 kWh, and the annual amount of electricity savings for farmers are USD 1896.4 if the unit rate of 0.041 USD/kWh is applied (Supplementary Table S1B). In the case of KEPCO, the annual benefit is estimated to be USD 1668.8 when the price gap between system marginal price and electricity price for agriculture is applied to the amount of electricity saved (Supplementary Table S1C). The annual greenhouse gas reduction from electricity savings for agricultural use is estimated to be 21.61 $tCO_2$ per year, and the annual greenhouse gas reduction benefit is estimated to be USD 394.8 if the emission permit price of 18.27 $USD/tCO_2$ is applied (Supplementary Table S1D). The annual economic benefit from solar power use is determined by the sum of the amount of annual cost reduction, benefits of annual reduction, and amount of reduction (Table 3). Therefore, the annual economic benefit from solar power of Bif A and Bif B use is estimated to be USD 3960. Based on the annual profits determined by cultivation area in 2015 and 2016 year, the annual broccoli production profits from the same cultivation area from solar power of Bif A and Bif B were estimated USD 379.2 (Table 4). The annual economic benefit from solar power was 10.4 times more than the broccoli production benefits. Therefore, farmer benefits will increase as they are cultivated in AV compared to open-field.

**Table 3.** The annual economic benefit from solar power use is determined by the sum of the amount of annual cost reduction, benefits of annual reduction, and the amount of reduction.

| Revenue | Amount |
|---|---|
| Benefits of annual cost reduction (USD) | 1896 |
| Annual cost reduction by AV (USD) | 1669 |
| co-benefits of carbon emissions reduction (USD) | 395 |
| Sum (USD) | 3960 |

**Table 4.** The annual broccoli production under BiF_A and B were used for annual profits determined by cultivation area.

| Revenue | Amount |
|---|---|
| Cultivation Area (a) | 3.24 |
| Profits per area (USD) | 116.9 |
| Annual broccoli profits under AV (USD) | 378.7 |

## 4. Conclusions

The importance of renewable energy is increasing not only in Korea but also around the world. In the case of Korea, the "Renewable Energy 3020 Implementation Plan" policy has been implemented to increase renewable energy. The agricultural sector has become interested in AV that produces electrical energy and food. In this study, microclimate, including PPFD and soil temperature, changed under AV, resulting in a small decrease in crop production and altered metabolites in broccoli. The additional shading in AV increased consumer preference for the product by improving its appearance quality. AV presents an opportunity to overcome the difficulties of cultivating in the open field due to severe weather phenomena and will make it possible to open a premium market by producing high-quality crops. The increased energy production will offer great advantages

to farmers. In terms of land use efficiency, AV is a good means of producing energy and food in Korea, which is a highly mountainous area.

**Supplementary Materials:** The following supporting information can be downloaded at: https://www.mdpi.com/article/10.3390/agronomy12061415/s1, Figure S1: Five glucosinolates of broccoli grown with or without solar panels. Figure S2: Six glucosinolate hydrolysis products of broccoli grown with or without solar panels. Figure S3: A graph showing a correlation between electricity generation and solar radiation. Table S1: Annual economic benefit calculation from solar power.

**Author Contributions:** Conceptualization, K.-M.K.; methodology, S.-H.C., and H.J.K.; software, S.-H.C., H.-W.M., and H.J.K.; validation, S.-H.C., H.-W.M., and H.J.K.; formal analysis, S.-H.C., H.J.K., Y.H.K.; writing—original draft preparation, S.-H.C., H.J.K., H.-W.M., and K.-M.K.; writing—review and editing, S.-H.C., H.J.K., and K.-M.K.; visualization, S.-H.C., H.J.K., and K.-M.K.; supervision, K.-M.K.; project administration, K.-M.K.; funding acquisition, K.-M.K. All authors have read and agreed to the published version of the manuscript.

**Funding:** This work was supported by the GS Construction and National Research Foundation of Korea (NRF) grant funded by the Korea government. (MSIT) (2021R1C1C1007733).

**Acknowledgments:** This research was funded by GS Construction. We thank those who helped with field preparation (tillage) work (Jeung Young Guen) and administration work at the Institute for Agricultural Practice Education, Chonnam National University in the Naju experiment station (team leader: Jin Kyoung Kim). The authors thank Su-mi Seo for helping graphical abstract.

**Conflicts of Interest:** The authors declare no conflict of interest.

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
