# Peer review of "Agrivoltaic Systems Enhance Farmers’ Profits through Broccoli Visual Quality and Electricity Production without Dramatic Changes in Yield, Antioxidant Capacity, and Glucosinolates"

_agronomy, doi:10.3390/agronomy12061415_

Round 1

Reviewer 1 Report

Overall, the topic of the manuscript could be of great interest within an evolving suite of research on agrivoltaic systems.  I have two main issues with the manuscript in its current form.  First, the entire manuscript could benefit from extensive editing of English language and style, which would improve the readability.  The second major issues is with the promise from the title "Agriphotovoltaics Produces Green Broccoli with Higher Consumers’
Preference under the Energy–Food–Water–Carbon Nexus".  The first portion of this title is supported by robust research - agrivoltaics resulted in more green broccoli heads.  The second portion of this title feels significantly under-supported.  In fact, the authors note that previous research from Topcu et al. and Garitta et al. reported that consumer was preferred green color of broccoli head. To support this research, this team simply posted a survey on a messenger board...  More specifically, "To compare the appearance of broccoli grown form control and APV, two pictures
which represent each treatment were sent to 57 men and 87 women by social network service messenger. People answered the question of which broccoli in the photo they were willing to buy."  This is not robust science. Again, with the calculation of the CO2 emissions or various energy sources (coal, oil, gas, and solar panel), this is simply using 4 different multipliers. There isn't robust work on the water dimensions either, so to say 'under the Energy–Food–Water–Carbon Nexus' feels very misleading.

This is unfortunate because the authors also do a great job analyzing the Total phenolic content (TPC), Total flavonoid content (TFC), antioxidant activity, and quantity of glucosinolate.  Why not focus on this more rigorous data when describing your paper?

Author Response

We changed title of manuscript as you suggested. Other comments were also helpful. We edited as you suggested.

Thank you.

Reviewer 2 Report

Dear Authors,

The research is well described and thought out. The influence of an AV system on the yield and quality of broccoli has not been described in such detail so far. The manuscript is packed with (too) many data, findings and observations. 

My recommendations for improvement are:

  • consider concentrating on agronomic findings and what is needed to understand them.
  • Introduction contains a lot of information on electricity problems in Korea and renewable energy generation. Please shorten this to max. one short paragraph.
  • Remove survey on visual quality. This may be intersting to present at a conference as a poster.
  • Are chapters 2.12 and 2.13 really necessary when talking about broccoli? I do not think so.
  • This means removing the results chapters 3.7 and 3.8.
  • Chapter 3.1: in my opinion, the majority of this paragraph belongs to material&methods section. Please reconsider to move parts of the paragraph.
  • Figure 3: the description mentions three seasons, but I understood it as three days in November 2020. Please check.

Author Response

Dear Reviewers,

Thank you for your insightful comments. 

We revised as you suggested.

Thank you.

Kang-Mo Ku

Round 2

Reviewer 1 Report

The manuscript is much improved in terms of it's focus, scope, and readability.

However, even after this revision, and especially in light of the revised text, I still find the title of the manuscript misleading given the primary findings, which are not the primary points of the research done here. The title feels more flashy and headline grabbing than the findings suggest, so I would not recommend publication unless the title better reflects the heart of the work (antioxidant capacity, and secondary metabolite of broccoli).

For example, the abstract notes that "The yield, antioxidant capacity, some glucosinolates and hydrolysis products of broccoli grown under an agrivoltaic system were not significantly different from those of broccoli grown in the open-field", yet the title states "Agrivoltaic Systems Enhance Broccoli Quality"

Author Response

Dear Reviewers,

Thank you for your insight.

We appriciate your suggestion and revised title as below.

"Agriphotovoltaics enhances famer's profits through broccoli visual quality and electronicity production without dramatic changes of yield, antioxidant capacity, and glucosinolates "

Thank you.

Kang-Mo Ku